

# Integrating multi-type aberrations from DNA and RNA through dynamic mapping gene space for subtype-specific breast cancer driver discovery

Jianing Xi[1], Zhen Deng[2], Yang Liu[1], Qian Wang[1] and Wen Shi[1]

[1] School of Biomedical Engineering, Guangzhou Medical University, Guangzhou, China
[2] School of Basic Medical Sciences, Guangzhou Medical University, Guangzhou, China

## ABSTRACT

Driver event discovery is a crucial demand for breast cancer diagnosis and therapy. In particular, discovering subtype-specificity of drivers can prompt the personalized biomarker discovery and precision treatment of cancer patients. Still, most of the existing computational driver discovery studies mainly exploit the information from DNA aberrations and gene interactions. Notably, cancer driver events would occur due to not only DNA aberrations but also RNA alternations, but integrating multi-type aberrations from both DNA and RNA is still a challenging task for breast cancer drivers. On the one hand, the data formats of different aberration types also differ from each other, known as data format incompatibility. On the other hand, different types of aberrations demonstrate distinct patterns across samples, known as aberration type heterogeneity. To promote the integrated analysis of subtype-specific breast cancer drivers, we design a "splicing-and-fusing" framework to address the issues of data format incompatibility and aberration type heterogeneity simultaneously. To overcome the data format incompatibility, the "splicing-step" employs a knowledge graph structure to connect multi-type aberrations from the DNA and RNA data into a unified formation. To tackle the aberration type heterogeneity, the "fusing-step" adopts a dynamic mapping gene space integration approach to represent the multi-type information by vectorized profiles. The experiments also demonstrate the advantages of our approach in both the integration of multi-type aberrations from DNA and RNA and the discovery of subtype-specific breast cancer drivers. In summary, our "splicing-and-fusing" framework with knowledge graph connection and dynamic mapping gene space fusion of multi-type aberrations data from DNA and RNA can successfully discover potential breast cancer drivers with subtype-specificity indication.

## INTRODUCTION

Breast cancer has become the most prevalent cancer in women worldwide, where 7.8 million women were diagnosed with breast cancer during the past 5 years at the end of 2020 (*Ferlay et al., 2021*). In cancer cells, driver events in genomics play important roles in

Corresponding author
Wen Shi, shiwen@gzhmu.edu.cn

tumorigenesis (*Bailey et al., 2018*), but the current understanding of breast cancer drivers, especially for the poor prognoses triple-negative subtype, is still limited (*Cancer Genome Atlas Network, 2012*). Consequently, driver event discovery is a crucial demand for breast cancer diagnosis and therapy. Thanks to the unprecedented achievements in DNA sequencing technology (*Nekrutenko & Taylor, 2012*), the cost of collecting large cohort of cancer genomic data reduces largely, leading to the opportunity for computational discovering cancer driver through the big data of breast cancer samples (*Ding et al., 2014*). In recent researches, numerical studies of computational driver discovery have emerged, but most of the existing studies only focus on the detection of whether the genes to be tested are cancer drivers, lacking of the indication of subtype-specificity (*Xi et al., 2020*). It should be noted that, indicating the subtype-specificity of drivers is an important aspect in precision medicine of cancers (*Alizadeh et al., 2015*). Actually, subtype-specificity of drivers can prompt the personalized biomarker discovery and precision treatment of cancer patients (*Cyll et al., 2017*). Therefore, building computational tools for the discovery of subtype-specific drivers is an urgent demand for the advancement researches in precision medicine of breast cancer.

Since the most relevant basis of cancer drivers is DNA aberrations in cancer samples, the existing computational driver detection approaches mainly focus on DNA sequencing data, and utilize the information of aberrations like single nucleotide variations (*Lawrence et al., 2013*; *Tamborero, Gonzalez-Perez & Lopez-Bigas, 2013*; *Luo et al., 2019*), copy number alternations (*Mermel et al., 2011*; *Xi & Li, 2015*), and structural variation (*Quigley et al., 2018*), *etc*. In consideration that genes have functional interactions with other genes, some recent approaches for driver discovery also expand the information of genomics from gene mutations to gene interactions (*Cowen et al., 2017*). For example, HotNet2 regards the gene interactions as a network with gene nodes (*Leiserson et al., 2015*), and propagates the mutations throughout the gene network to integrate the information of mutations and interactions of genes. Subsequently, by restricting propagating step length (*Cho et al., 2016*) or diffusion scale (*Babaei et al., 2013*), many revisions of network propagation for genomic data integration are also proposed to alleviate the false positives from unrestricted propagation (*Cowen et al., 2017*). DawnRank is designed to discover personalized drivers of a single patient by perturbation ranking on the interaction network (*Hou & Ma, 2014*). These interaction based driver methods have been widely-accepted as computation tools for driver discovery in recent researches (*Cowen et al., 2017*), and they mainly exploit the information from DNA aberrations and gene interactions.

It should be noted that, cancer driver events would occur due to not only DNA aberrations but also RNA alternations (*Calabrese et al., 2020*). For example, the PCAWG Transcriptome Core Group has systemically characterized tumor transcriptomes from samples of more than thousands of donors and several cancer genomic databases, and comprehensively analyze the catalogue of cancer-associated RNA alterations of genes (*Calabrese et al., 2020*). They also observe the abundance of co-occurrences of RNA and DNA alterations and recurrent RNA alterations in driver genes (*Calabrese et al., 2020*). However, there are few computation tools of cancer driver identification take into account the RNA alternations (*Calabrese et al., 2020*). As demonstrated by the PCAWG

Transcriptome Core Group, there are also correlations between the DNA and RNA alternations, and these correlations can connect the two types of aberration data to realize the integration of DNA and RNA aberrations (*Calabrese et al., 2020*). Unfortunately, most existing cancer driver detection methods underestimate the information in RNA alternations, only exploit the information from either genomic aberrations or gene interactions. The cancer driver discovery methodology for the integration analysis of the multiple types of aberrations from both DNA and RNA is still lacking.

Nevertheless, for breast cancer driver analysis, integrating multi-type aberrations from both DNA and RNA is a challenging task. On the one hand, the data formats of different aberration types also differ from each other, known as data format incompatibility. Since storing the data into a unified storage is a prerequisite of data integration, the mutations of DNA and altered expressions of RNA are two incompatible formats to each other (*Cancer Genome Atlas Network, 2012*). On the other hand, different types of aberrations demonstrate distinct patterns across samples, known as aberration type heterogeneity (*Calabrese et al., 2020*). Specifically, different aberration types show mismatched the statistical distributions (*Pereira et al., 2016*), which are difficult to be integrated as one single statistical variable. A widely-used compromise strategy is measuring each type of aberrations into type-specific similarities, and integrating the series of type-specific similarities by weighted summation (*Wang & Ma, 2022*; *Xiao et al., 2020b*, *2020a*) or regularization (*Xi, Li & Wang, 2018*). But there is still a drawback for similarity-based integrations, since they would cause the degradation of multi-type aberrations, leading to only a scalar result with deficiency of the informative multi-dimension aberrations (*Xi, Li & Wang, 2018*). Therefore, how to integrate aberrations from both DNA and RNA into a unified format with the informative multi-type aberration heterogeneity preserving, is still a bottleneck for driver discovery from multi-type aberrations.

To promote the integrated analysis of subtype-specific breast cancer drivers, we design a "splicing-and-fusing" framework to address the issues of data format incompatibility and aberration type heterogeneity respectively. In the "splicing-step" of our framework, we firstly adopt knowledge graph structure to reformat the DNA and RNA data into a unified formation of multi-type aberrations (*Nickel et al., 2015*), which can successfully overcome the data format incompatibility by connecting the information from different sources as a series of triplets of facts. Furthermore, we also propose a dynamic mapping (*Ji et al., 2015*) gene space integration approach in the "fusing-step". In consideration of the aberration type heterogeneity, this gene space approach can represent the multi-type information into a vectorized profile, instead of scalar representation (*Wang et al., 2017*). To evaluate the efficiency of our integration analysis approach for subtype-specific drivers on breast cancer data (*Cancer Genome Atlas Network, 2012*), we conduct experiments of comparison study to assess the cancer driver discovery performance, and ablation studies on both data and methods to assess the effects of multi-type information and dynamic mapping strategy. We also employ experiment of data visualization for subtype-specificities indication of the discovered potential drivers, and further unscrambling gene functions by enrichment analysis experiment. The experimental results indicate the superiority of our approach in the discovery of subtype-specific breast cancer driver, and the advantage of integrating

multi-type aberrations from both DNA and RNA for cancer drivers. Our code files are publicly offered for more researchers to use the model to calculate the drivers. We have provided the source code of our model on GitHub: https://github.com/JianingXi/DynMap-Integrating-multi-type-aberrations.

## MATERIALS AND METHODS

### Data acquisition of breast cancer samples

The data of DNA and RNA aberrations of breast cancer samples are collected from The Cancer Genome Atlas (TCGA) project, a well-curated database including DNA sequencing data and RNA expression data of cancer samples (*Cancer Genome Atlas Network, 2012*). The cancer samples with both DNA and RNA data available are selected as our integration analysis, where the total volume of 523 breast cancer samples. To simply the preprocessing of the TCGA data, we adopt a pre-compiled source, of the TCGA breast cancer data from the UCSC Xena platform for cancer genomics data (*Goldman et al., 2019*). The UCSC Xena platform provides the occurrence of gene aberrations in DNA of each sample and the abundance of gene expressions in RNA of each sample (*Goldman et al., 2019*). Specifically, there are several different types the occurrence of gene aberrations in DNA, including 3′ Flank, 3′ UTR, 5′ Flank, 5′ UTR, frame shift del, frame shift ins, IGR, in frame del, in frame ins, intron, missense mutation, nonsense mutation, nonstop mutation, silent, splice region, splice site, and translation start site. At the same time, the abundance of gene expressions in RNA is also utilized in mining differentially expressed RNAs of genes (*Zhao et al., 2015*), including over-expressions and under-expressions. These differentially expressed RNAs of genes are regarded as RNA alternations of genes. Accordingly, the aberration data in DNA and RNA are aligned to their corresponding breast cancer samples, and the samples can play the roles as the anchor between the two distinct types of aberrations. The collected aberrations from both DNA and RNA data of each sample are the basis of data integration.

In addition to aberration data in DNA and RNA, we can also introduce side information to prompt the subtype-specific breast cancer driver discovery task in our integration analysis. For subtype-specificity indication task in our integration, the TCGA database also provides the subtype annotations of the breast cancer samples, including five widely-accepted intrinsic molecular subtypes: Luminal A, Luminal B, HER2-enriched, Triple-negative, and Normal-like subtypes (*Cancer Genome Atlas Network, 2012*). The subtype annotations of the breast cancer samples can also be obtained from the UCSC Xena platform (*Goldman et al., 2019*). Here the subtype annotations can play the role as side information for subtype-specificity indication of the investigated genes. Furthermore, in consideration that the gene interactions also play important roles in tumorigenesis (*Cowen et al., 2017*), accordingly we also incorporate the gene interaction data into our integration analysis as network format data, regarding the genes as network nodes and the interactions as network edges (*Leiserson et al., 2015*). The gene interaction information is collected from two classical databases, STRING (*Franceschini et al., 2012*) and iRefIndex (*Razick, Magklaras & Donaldson, 2008*), facilitating the customizing integration with different versions of curated interactions. The annotations of genes being known cancer

drivers are also introduced as the supervision information to support the cancer driver discovery in the integration analysis. In summary, we include gene aberrations from DNA, abnormalities from RNA, subtype annotations of samples, gene interactions, and known cancer driver annotations of genes as the data used in the integration analysis (Fig. 1A).

## Overview of "splicing-and-fusing" framework

Since the RNA alternation and DNA mutations are distinct types of aberrations of genes, their integration faces the challenges of data format incompatibility and aberration type heterogeneity. Therefore, we design a "splicing-and-fusing" framework to address the two challenges step by step. After the data collection (Fig. 1A), we first feed the data into "splicing-step". To overcome data format incompatibility issue, in the "splicing-step" the multi-type data from DNA, RNA and other sources are redescribed as a series of facts of triplets, and the data of facts are spliced together by connections across the triplets (Fig. 1B). Therefore, the "splicing-step" can successfully address the problem of data format incompatibility and joint the multi-type data into a flexible and unified format as knowledge graph. The next step of our "splicing-and-fusing" framework is the "fusing-step" (Fig. 1C). In this step, we introduce the data representation of knowledge graph and adopt a dynamic mapping gene space fusing strategy. The idea of dynamic mapping can prompt the relation-specific resolution of data fusion for many-sided representations, and the gene space representations can describe the multi-type information into vectorized profiles. Compared with scalar score, our vectorized gene space can offer an informative many-sided representation to describe the aberration type heterogeneity. After the dynamic mapping gene space representations are obtained, we further apply the indication of subtype-specific cancer drivers in "discovering-step" (Fig. 1D). Here we use the recovery loss score learnt during "fusing-step" to infer whether the investigated genes are a driver or not and indicate which subtype the drivers would belong to. Finally, we can obtain the breast cancer drivers from the investigated genes as well as their subtype-specificities.

## "Splicing-step": connecting data format incompatibility

To address the data format incompatibility of multi-type aberrations, we convert the data of each type as a series of itemized facts (*Nickel et al., 2015*), and splice these facts as a knowledge graph. Specifically, for both DNA and RNA, we can record a fact as a triplet (sample $P$, aberration type $T$, gene $G$) if gene $G$ occurs with a $G$ type aberration in sample $T$ (Fig. 1B). Here a triplet fact includes three elements: subject, predicate, and object. Here the subject and object are the head $h$ and tail $t$ of the triplet fact, respectively, and the predicate is the relation $r$ between the head $h$ and tail $t$. When we list all the triplet facts of aberrations from both DNA and RNA, every type of aberrations in their related genes of all the samples are fully covered in these facts. Here the subjects include all samples and the objects include all aberrated genes, which are regarded as entities in the triplets.

The aberration types between the samples and genes are regarded as relations in the triplets. When we collect all the entities as graph nodes and relation as graph edges, the connections between the entities and relations can be formed as a graph structure with these different types of nodes and edges as a knowledge graph (*Nickel et al., 2015*). Since

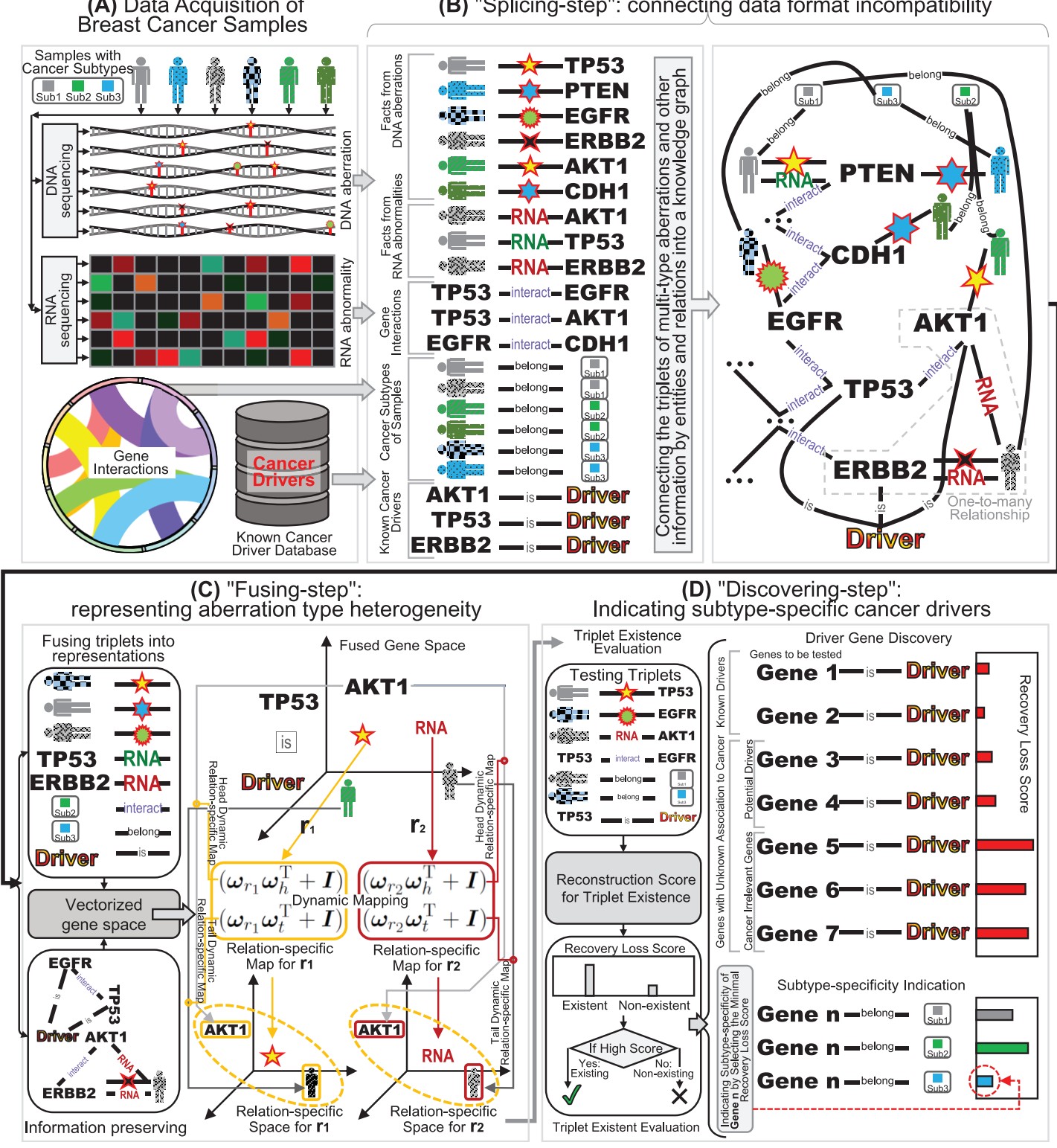

**Fig. 1** The schematic diagram of our "splicing-and-fusing" framework for subtype-specific cancer driver discovery. (A) Data acquisitions of DNA and RNA aberrations data from breast cancer samples, gene interactions, and known benchmarking drivers. (B) The "splicing-step" connecting data format incompatibility *via* constructing the knowledge graph. (C) The "fusing-step" representing aberration type heterogeneity *via* dynamic mapping relation-specific gene space. (D) The "discovering-step" indicating subtype-specific cancer drivers *via* recovery loss scores.

Full-size 🖾 DOI: 10.1093/bioinformatics/btu811

the edges in the knowledge graph can preserve different types of relations, this connection structure can overcome the incompatibility of different data formats of aberrations from DNA and RNA.

Since the information related to breast cancer drivers is far beyond the aberrations of DNA and RNA, such as the interactions between genes and the subtype annotations of breast cancer samples (*Cowen et al., 2017*), we also include the side information into the connection structure of knowledge graph. Specifically, the knowledge graph structure can also incorporate gene interactions by adding the triplet facts of gene interaction relationships. Here we can format a triplet (gene $G_1$, interaction $I$, gene $G_2$) if there exists an interaction between gene $G_1$ and $G_2$ (Fig. 1B). Then the facts of gene interactions can be easily included in the knowledge graph by only listing additional triplets related to interaction. Note that many genes may have several synonym names (*Sherman et al., 2022*), we can also list the synonym facts (gene $G_1$, synonym $S$, gene $G_2$) and feed these triplets into the knowledge graph. When all the triplet facts are listed together, we can obtain the entity set of all the heads (subjects) and tails (objects), and the relation set of all the predicates (multi-type aberrations between samples and genes, and interactions/synonyms between genes). Another type of side information is the triplets related to subtypes of breast cancer (*Cancer Genome Atlas Network, 2012*). These subtype annotations of samples can be formed as triplets (sample $P$, belong to $B$, subtype $S$), when the investigated breast cancer sample $P$ (subject) belongs to (predicate) the certain subtype $S$ (object) (Fig. 1B). The inclusion of subtype annotations plays the key role in the subtype-specificity indication in the next subsection.

Finally, we also add the facts of the existing known cancer drivers (*Sondka et al., 2018*) by triplets (gene $G$, is $Is$, driver $D$) (Fig. 1B). These facts also play the roles of supervision information in the constructed knowledge graph. Note that it is still unknown whether many aberrant genes are associated with cancer or not, but these genes are included in the triplets as entities due to their participation in genomic aberrations or gene interactions (*Sondka et al., 2018*). Accordingly, the collected triplets of the facts that some genes are known cancer drivers can only cover a subset of driver genes, and some other genes in the triplets are potential drivers to be discovered. Here a collected triplet with cancer driver annotation (gene $G$, is $Is$, driver $D$) can indicate that the gene $G$ in this triplet is an experimental validated known cancer driver, but the absence of a driver annotation triplet of a gene cannot indicate its irrelevance to cancer driver (*Wang et al., 2017*; *Xi, Miao & Huang, 2021*). Actually, the absence of driver annotation can only tell that we do not know whether the gene is driver or not. Through the collections of triplets aforementioned, when we regard the same entities and relations across these triplets as graph nodes, we can connect the triplets of facts as graph edges (Fig. 1B). The connection result can form an integrated knowledge graph including a series of triplet facts of multiple types of aberrations from both DNA and RNA. Since this step contains the operation of splicing together different triplet facts into an integrated knowledge graph, we denote this multi-type aberration data connection step as "splicing-step", and this knowledge graph based splicing-step *via* can successfully overcome the data format incompatibility problem.
### "Fusing-step": representing aberration type heterogeneity

To preserve the aberration type heterogeneity in the integrated representation, we further propose a dynamic mapping gene space integration approach, known as the "fusing-step". Unlike the previous scalar based methods such as statistical variation approaches (*Lawrence et al., 2013*; *Tamborero, Gonzalez-Perez & Lopez-Bigas, 2013*) or similarity based integration approaches (*Leiserson et al., 2015*; *Cho et al., 2016*), we select a vectorized profile containing multiple dimensions to represent the information of genes with multi-type aberrations. In the vectorized profiles of gene representations, each gene denotes a multi-dimension vector that can contain various trends across different samples in a mapped multi-dimension gene space. Fortunately, there are several advanced methods that can support the multi-dimension vector representations of knowledge graph, throwing light on the learnt space representations of multi-type aberrations (*Wang et al., 2017*). Inspired by the representation learning advancements, we can fuse the knowledge graph containing multiple types of aberrations into an integrated gene space with the graph structure preserved, known as knowledge graph embedding (*Wang et al., 2017*). Thus, in this article, we propose a dynamic mapping strategy to embed the knowledge graph into integrated gene space for fusing the DNA and RNA aberrations.

Technically, the learnt fused representations are expected to successfully recover the raw triplets in the multi-type aberration knowledge graph, *i.e.*, the fusing strategy should be information preserving (*Nickel et al., 2015*). At the same time, the learnt fused representation of the knowledge graph should also be in a low-dimensional space (*Wang et al., 2017*). This is due to the hypothesis that the lower the learnt gene space dimension, the more concise the computational discovery would be (*Wang et al., 2017*). Consequently, fusing the aberration information can be formulated as a task that using a series of vectors in a learnt space to approximately recover the known triplets in knowledge graph (*Wang et al., 2017*). Specifically, a common strategy for triplet information fusion is to represent the relationships of entities and relations in the triplets through translations operating on the low-dimensional representations (*Wang et al., 2017*; *Bordes et al., 2013*). By optimizing the translations between the head and tail on a certain relation through a distance metric score as $Dis^r(h, t)$, we can obtain a series of vectors for entities and relations fusing the information in a representation space of genes.

For putting the idea of information fusion by triplet translations operating on the low-dimensional representations into practice, a straightforward but effective approach for learning a translation representation space of fused triplets is to minimize the residual of head and tail vectors by summation (*Bordes et al., 2013*). Specifically, the specific translation for a certain triplet $(h, r, t)$ can be modeled as a vector summation in the representation space as $v_h + v_r \approx v_t$. This approximation is expected to induce the information preserving of the learnt representation vectors (also known as embedding vectors) of the entities and relations in the knowledge graph (*Bordes et al., 2013*). A common choice is to minimize the L1-norm on the vector residual $v_h + v_r - v_t$, formed as the distance between the head entity $h$ and tail entity $t$ on a certain relation $r$ in the fused triplet $(h, t, r)$:

$$Dis^{(r)}(v_h, v_t) = \|v_h + v_r - v_t\|_1, \tag{1}$$

where the subscript 1 denotes that the metric is set to L1-norm in consideration of its sparse penalty property in the role of distance (*Bordes et al., 2013*). The summation based fusing method, also known as TransE (*Bordes et al., 2013*), can provide the representations as translations in the fused vector space.

Despite the efficiency of the simple translational fusing model (*Bordes et al., 2013*), a shortcoming for the summation based translational fusing approach is that this method cannot distinguish the representations for one-to-many triplets, thanks to the uniqueness of summation (*Wang et al., 2014*). However, in a breast cancer sample, different types of aberrations are likely to occur in one certain gene at the same time, and this phenomenon shows that one-to-many triplets are inevitable in multi-type aberration knowledge graph data (*Wang et al., 2014*). Consequently, the representations of entities in gene space should be many-sided when confronting multi-type aberrations as multiple relations.

In consideration that different entities in one-to-many triplets are usually associated with distinct relations, a plausible solution is projecting the representation vectors to different relation-specific hyperplanes (also known as mapping on planes), so that the entity vector can be represented as many-sided in fused gene space (*Wang et al., 2014*). Specifically, when an entity in one-to-many triplet confronts various relations, the information preserving of summation approximation can be revised as a summation of the projected vectors on relation-specific hyperplanes rather than a simple vector summation in the gene space:

$$Dis^{(r)}(v_h^{proj}, v_t^{proj}) = \left\| (v_h - \omega_r^T v_h \omega_r) + v_r - (v_t - \omega_r^T v_t \omega_r) \right\|_1, \tag{2}$$

where the vector $\omega_r$ is the normal vector of a relation-specific hyperplane, and $v_h^{proj} = (v_h - \omega_r^T v_h \omega_r)$ and $v_t^{proj} = (v_t - \omega_r^T v_t \omega_r)$ are the projection vectors of representations of head and tail entities in the fused gene space. This translating on hyperplanes fusing strategy, also known as transH (*Wang et al., 2014*), is a breakthrough to tackle multi-relation data. Through the relation-specific hyperplane projection in gene space, we can circumvent the multiple relations in one-to-many triplets in the learnt fused information.

To further enhance the information preserving of data fusing for multi-type aberration knowledge graph with one-to-many or many-to-many triplets, we can also introduce a relation-specific space instead of a hyperplane (*Lin et al., 2015*; *Ji et al., 2015*).

In comparison with hyperplanes, relation-specific spaces can cover more fine-grained connotations for entities when their representation vectors are mapped into various situations in one-to-many or many-to-many triplets (*Lin et al., 2015*). An intuitive solution to construct a relation-specific space is to build a static map from the fused representation gene space to the relation-specific space (*Wang et al., 2017*). Specifically, the coefficients of the static map can be described as a series of elements in a relation-specific matrix $M_r$, and the linear transformation can be constructed as a matrix product such as $v_h^{StaticMap} = M_r v_h$ and $v_t^{StaticMap} = M_r v_t$ for heads and tails respectively (*Lin et al., 2015*). Under the idea of relation-specific spaces, the information preserving of approximation is the distances between the head and tail entities on the relation-specific spaces are formulated as:

$$Dis^{(r)}(v_h^{StaticMap}, v_t^{StaticMap}) = \|(M_r v_h) + v_r - (M_r v_t)\|_1. \tag{3}$$

This idea of information fusion by translation in the corresponding relation space is also known as TransR (*Lin et al., 2015*). However, when the number of relation types is large, the degree of freedom of the parameters in relation space map in the relation-specific matrix also grows rapidly, causing the underdetermined problem and overfitting risk (*Xi et al., 2021b*).

To reach a higher relation-specific resolution data fusion and a dynamic many-sided representation of the head and tail of triplet data, we further introduce the idea of dynamic mapping matrix (*Ji et al., 2015*) into the relation-specific space fusing multi-type aberration knowledge graph. Rather than using a relation-specific matrix with full degree-of-freedom static parameters, we choose a compression scheme (*Ji et al., 2015*) to build a relation-specific space with a mapping matrix that are dynamic for the different entities and relations (Fig. 1C). For each triplet in multi-type aberration knowledge graph, the parameter matrix of the dynamic mapping is computed by projection vectors of both entity and relation. To ensure fine-grained fusion representations in the multi-type aberration triplets (*Ji et al., 2015*), the projections are also dynamic for the entities between heads and tails as $M_r^h = \omega_r \omega_h^T + I$ and $M_r^t = \omega_r \omega_t^T + I$ respectively. Accordingly, the information preserving of approximation for dynamic mapping can simultaneously incorporate the diversity of entities and relations (Fig. 1C):

$$Dis^{(r)}(v_h^{DynMap}, v_t^{DynMap}) = \left\|[(\omega_r \omega_h^T + I)v_h] + v_r - [(\omega_r \omega_t^T + I)v_t]\right\|_1, \tag{4}$$

where $v_h^{DynMap} = M_r^h v_h$ and $v_t^{DynMap} = M_r^t v_t$. To fuse the information in the triplets of the knowledge graph, we can perform the Adam optimizer (*Kingma & Ba, 2015*) on the joint loss function for recovering all the existing triplets of facts:

$$\min_{(v_h, v_r, v_t, \omega_h, \omega_r, \omega_h)} \sum_{(h,r,t) \in G} Dis^{(r)}(v_h^{DynMap}, v_t^{DynMap}), \tag{5}$$

where the symbol $G$ denotes the set containing all the triplets of facts in the integrated knowledge graph. Finally, after the information preserving representations are learnt *via* optimizing the aforementioned function, the one-to-many and many-to-many triplets with multiple relations can dynamically represent the multi-type aberration knowledge graph through the dynamic mapping gene space (Fig. 1C).

## "Discovering-step": indicating subtype-specific cancer drivers

It should be noted that the motivation of integrating multi-type aberrations from DNA and RNA is not for data integration itself, but for the clinical application of subtype-specific breast cancer driver discovery. Due to the fact that all the genes in genomics are contained in the aberration knowledge graph in the "splicing-step" of the integration, both the known cancer driver genes and the genes with unknown association with cancer are all included in the graph (*Xi, Miao & Huang, 2021*). Consequently, the task of cancer driver discovery can be equivalent to disclosing the potential driver genes from the genes with unknown association to cancer, since there are also cancer irrelevant genes

included. Although the genes to be tested are already included in the multi-type aberration knowledge graph, the entities of the potential driver genes do not have known associations with the cancer driver entity, and hence the knowledge graph in established "splicing-step" cannot discover the potential cancer drivers directly. Fortunately, in the "fusing-step", the fused representations in the gene space of multi-type aberration knowledge graph can be applied in cancer driver discovery, thanks to the link prediction application of graph embedding (*Rossi et al., 2021*; *Xi et al., 2021b*). In the "fusing-step", the fused representations in the gene space not only can recover the known triplets of facts in the multi-type aberration knowledge graph, but also can discover the potential driver genes of breast cancer.

For the cancer driver discovery from the dynamic mapping gene space, the reason for utilizing the fused information of the gene entities is that their representations in dynamic mapping gene space can recover or predict their relations with driver entity. Since the fusing process not only optimize the recovery loss of the existing known triplets but not the unknown triplets, the potential relations between the genes and the entity of cancer driver are not restricted to be negative (*Wang et al., 2017*; *Xi, Miao & Huang, 2021*), but also feasible to be positive if the genes are potential drivers (*Xi et al., 2022*; *Xi, Miao & Huang, 2021*). Consequently, by computing the recovery loss function on the genes to be tested (*Rossi et al., 2021*), we can obtain the loss score $Dis^r(h, t)$ of these genes across the triplets with driver entity (Fig. 1D), where the tested triplet $(h, r, t)$ is $(h, r, t) =$ (gene to be tested $G_{test}$, is $Is$, driver $D$). If the loss score $Dis^{Is}(G_{test}, D)$ is smaller, then the tested gene $G_{test}$ has more potential to be a cancer driver. The basis that the recovery loss score has the capability of reflecting cancer drivers is the inclusion of known drivers as supervision information in the knowledge graph data (*Luo et al., 2019*). Thus, the potential driver genes tend to have small values of the recovery loss scores $Dis^{Is}(G_{test}, D)$ in triplets (gene to be tested $G_{test}$, is $Is$, driver $D$). The recovery loss scores are inducted to yield small values for all the triplets of known drivers in the knowledge graph data, and therefore the scores of potential cancer driver genes are also expected to be small for unobserved triplets including the genes to be tested (*Luo et al., 2019*). Consequently, we can apply the fused data representation from multi-type aberration knowledge graph on the discovery of potential breast cancer drivers.

Since only inferring whether a gene to be tested is a potential driver is not enough for the indication of subtype-specific drivers for breast cancer, therefore we should also indicate the subtype-specificity of the discovered driver genes (*Xi et al., 2020*). Note that there are no triplets of facts describing the indication relationships of driver genes to cancer subtypes as the form (gene $G$, belong to $B$, subtype $S$), But we can borrow the known information from the subtype belongingness of samples with the triplet form (sample $S$, belong to $B$, subtype $S$). Here the fused representations in gene space are expected to reflect the subtype information due to the inclusion of subtype related triplets in knowledge graph. Like the idea of cancer driver discovery above, we can also adopt the recovery loss scores to indicate the subtype-specificity of the investigated gene. For a potential driver gene $G$ discovered by the fused representations, we further utilize the recover loss score function $Dis^{\cdot}(\cdot, \cdot)$ on the unknown triplet (gene $G$, belong to $B$, subtype $S$). Accordingly, we

can yield the scores of the investigated gene $G$ across all the subtypes $S_1, S_2, \ldots, S_K$, as $Dis^B(G, S_1), Dis^B(G, S_2), \ldots, Dis^B(G, S_K)$. Since a small score indicates a better change of the unknown triplet, we can compare the scores across different subtypes, and assign subtype for the smallest score to the investigated gene as its subtype-specificity (Fig. 1D). Based on the recovery loss score assignment, we can achieve the subtype-specificity indication of cancer drivers through the fused representation gene space.

# EXPERIMENTAL RESULTS

## Experiment setup

### Experiment design

To examine the effectiveness for subtype-specific cancer driver discovery of our multi-type aberration integration approach, we conduct several experiments to analyze our approach from multiple perspectives: (1) comparing our approach with widely-used existing cancer driver approaches, (2) testing the effect of integrating multi-type aberrations by data ablation, (3) applying ablation study for gene space by removing dynamic mapping strategies, (4) indicating subtype-specificities of genes by data visualization, and (5) unscrambling gene functions by enrichment analysis. The first three experiments are conducted for examining the cancer driver discovery power from the gold standards of known benchmarking driver genes, in order to investigate the advancement of our approach, the effect of different type information, and the effect of data fusing strategies. Specifically, we compare our approach with existing interaction based driver methods including HotNet2 (*Leiserson et al., 2015*), DawnRank (*Hou & Ma, 2014*), two versions of MUFFINN (DNsum version and DNmax version respectively) (*Cho et al., 2016*), and our previous method DriverSub (including settings of k = 3 and k = 4) (*Xi et al., 2020*). The fourth experiment is performed to visually demonstrate the indication of subtype-specificities of drivers. The fifth experiment is applied to unscramble the gene functions of discovered drivers.

To provide the experiment design details of our model in driver identification task, here we also demonstrate two aspects on our used breast cancer samples. (1) The first aspect is collecting breast sample data containing both DNA and RNA aberrations. Regarding the first aspect, the reason of using DNA and RNA aberrations contained in the same samples is that we can investigate the integration of two types of aberrations only if both DNA and RNA aberrations are collected from the same samples. With the integration of two types of aberrations, our model can achieve a better performance on driver gene identification. (2) The second aspect is splitting the same cohort of breast cancer samples into two independent datasets. To avoid the ideal situation that all the used breast cancer samples in model training and performance testing are all the same from the TCGA database, we adopt the five-fold cross validation so that the testing breast cancer samples are separated from the dataset before the model training process. Through the five-fold cross validation, one-fifth of the breast cancer data are excluded from the model training stage, and these data only serve as the performance testing. Thus, although all the breast cancer data are

collected from the same TCGA database, we can successfully avoid the situation that the same sample is used in driver identification.

### Evaluation metrics

In the evaluation of cancer driver discovery, there are gold standards of known benchmarking driver genes, and consequently we easily examine whether a discovered gene is truly a cancer driver. Note that during the "splicing-step" of our approach, the information of the known cancer driver is a part of triplets of the facts include in the integrated knowledge graph as the supervision information. Thus, we employ the idea of cross validation for cancer driver discovery performance evaluation (*Luo et al., 2019*). Specifically, we evenly split the known cancer driver triplets into five folds, and select one of the five folds of triplets out of the integrated knowledge graph as testing set.
The remaining four folds of driver triplets are included in the integrated data as training set. We further selected the other folds as testing sets and keep the remains for training, and all the five folds can serve as five-fold cross validation. Since the comparison including both supervised methods and unsupervised methods, the outputs of genes to be tested in each fold are concatenated together so that the concatenated outputs of every gene are free from supervision information (*Xi et al., 2021b*). Accordingly, the outputs of genes are evaluated by receiver operating characteristic (ROC) curves, where the x-axis of the curve is the false positive rate (FPR) which is the fraction of discover genes in the non-benchmarking drivers (*Sondka et al., 2018*), and the y-axis of the curve is the true positive rate (TPR) which is the fraction of discover genes in the benchmarking drivers. As for visualization for subtype-specificity and enrichment analysis for unscrambling functions, there are no quantitative evaluation metrics, and we focus on the illustration of the results in these two experiments.

### Results of driver discovery comparison

To demonstrate the cancer driver discovery performance of our dynamic mapping gene space integrated multi-type aberration information, we adopt our approach on the TCGA breast cancer data (*Cancer Genome Atlas Network, 2012*) by constructing a knowledge graph and fusing the multi-type information to yield the recovery loss scores of the investigated genes. We also compare the widely-used existing interaction based driver approach HotNet2 (*Leiserson et al., 2015*), DawnRank (*Hou & Ma, 2014*), MUFFINN (*Cho et al., 2016*), and our previous study DriverSub (*Xi et al., 2020*) on the breast cancer dataset. Since we have two choices of gene interaction resources, STRING (*Franceschini et al., 2012*) and iRefIndex (*Razick, Magklaras & Donaldson, 2008*), here we demonstrate the performance of these competing methods under the inclusion of the two interaction resources separately. Through the ROC curve evaluation, we can observe that our approach achieves the best discovery performance of those of the competing methods. Specifically, when using the interaction resource of STRING, we can observe that DriverSub and DawnRank achieve better performance than those of HotNet2 and the two versions MUFFINN (Fig. 2A). Accordingly, the AUC values of DriverSub (k = 3), DriverSub (k = 4), and DawnRank are located in the range from 70.0% to 80.0% (Fig. 2B).

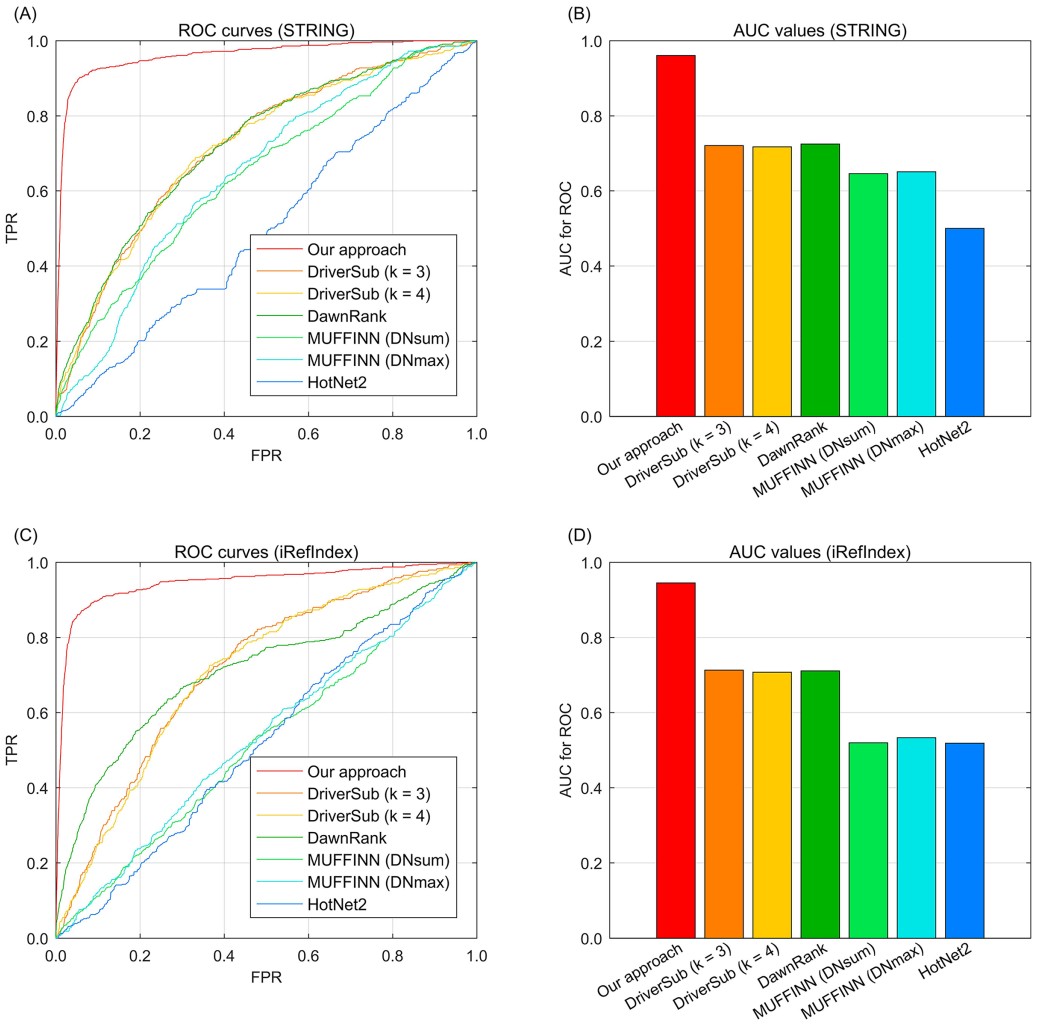

**Fig. 2** The ROC curves of our approach against those of the existing methods on breast cancer data and gene interactions. (A) ROC curves of the competing methods with interaction source of STRING. (B) AUC values of the competing methods with interaction source of STRING. (C) ROC curves of the competing methods with interaction source of iRefIndex. (D) AUC values of the competing methods with interaction source of iRefIndex.

Full-size 🖼 DOI: 10.1093/bioinformatics/btu811

In comparison, the curve of our approach is closest to the top-left corner (Fig. 2A), yielding an AUC value of 96.1%. As for the case of interaction resource of iRefIndex, we can observe a similar phenomenon that the curve of our approach fully covers the areas of the curves of the other competing methods (Figs. 2C–2D). A plausible explanation of the phenomena is that in comparison with the other existing methods, our approach integrates multi-types of information beyond the information of only DNA aberrations and gene interactions. Generally, our approach demonstrates a favorable performance on the task of cancer driver discovery.

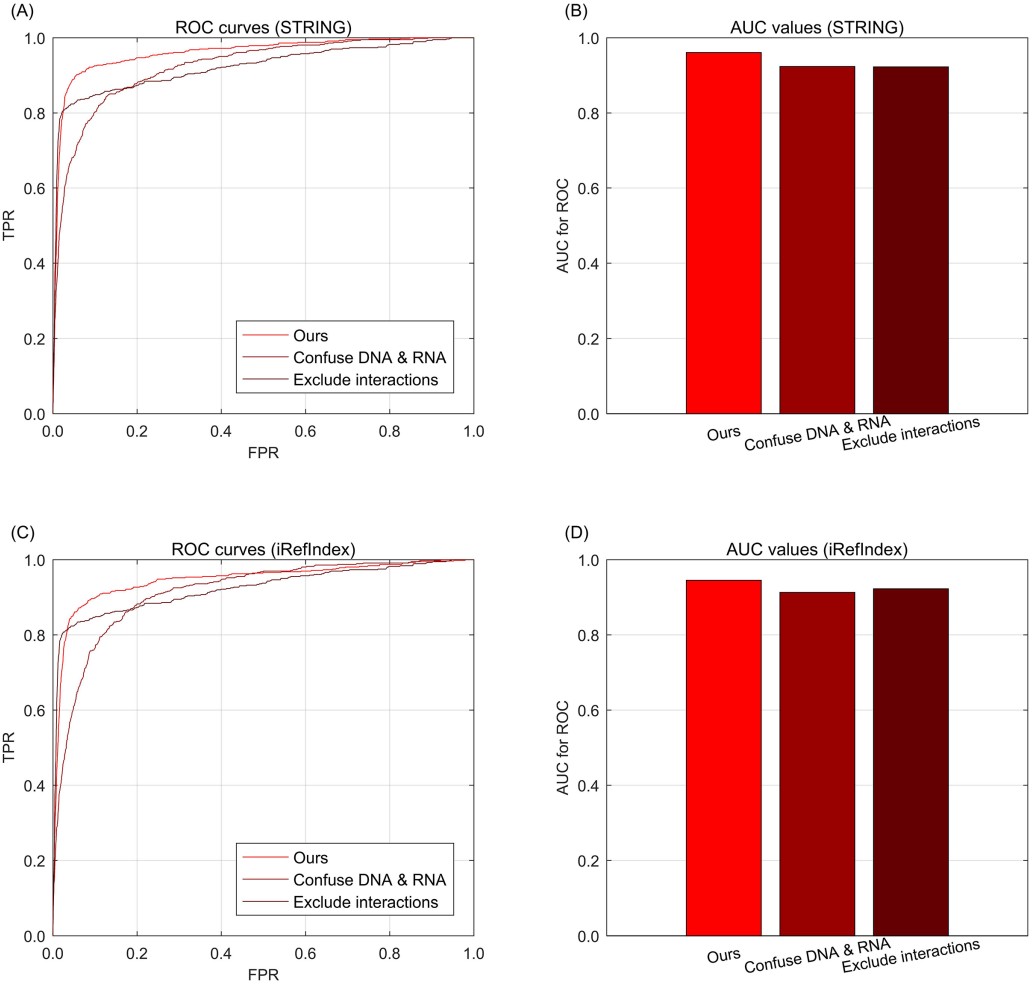

**Fig. 3 The ROC curves of our approach for data ablation on breast cancer data and gene interactions.** (A) ROC curves for data ablation with interaction source of STRING. (B) AUC values for data ablation with interaction source of STRING. (C) ROC curves for data ablation with interaction source of iRefIndex. (D) AUC values for data ablation with interaction source of iRefIndex.                                     Full-size ⬛ DOI: 10.1093/bioinformatics/btu811

## Results of multi-type aberration ablation

To further investigate the effects of information integrations from multi-type data, we further conduct the data ablation on the multi-type data. Here, data ablation is a strategy for observing the performance effects when the data are changed (*Mousavi, Khanal & Estrada, 2020*). When we confuse the multi-type aberrations from DNA and RNA as one single type, *i.e.*, only considering whether the genes are with aberrations but excluding the difference of aberration types between DNA and RNA. In the knowledge graph, the relations of different aberration types are confused as only one type of aberration, but the other types of relations such as interaction or subtype belongingness are not confused. As shown in Fig. 3A for STRING and Fig. 3C for iRefIndex, the curve of confusing multi-types is lower than that of the full version data of our approach, *i.e.*, no aberration type confusing. These phenomena indicate that the difference between multi-type

aberrations are informative for the driver discovery of our integration approach. Another way to change the integrated data is to exclude the information of gene interactions from the knowledge graph. By this way, we can observe the effects of performance from the interaction information. Through Figs. 3A and 3C, we can find that the performance decrease by excluding interactions are comparable with the decrease of confusing aberration types. Here the two curves in Figs. 3A and 3C are exactly the same due to their knowledge graphs are also exactly the same when the interactions are removed. The AUC values of the curves of the cases in data ablation are also provided in Figs. 3B and 3D for interaction resource with STRING or iRefIndex respectively.

## Results of gene space mapping ablation

In addition to data ablation that excluding part of the multi-type data, we also conduct the ablation study for gene space mapping by excluding part of the strategy during the knowledge graph representation. In our approach, the relation-specific space mapping is based on a dynamic strategy (*Ji et al., 2015*), denoted as DynMap in the ablation study (the full version of our approach). An ablation version of our approach is replacing the relation-specific dynamic mapping of with a relation-specific static mapping (*Lin et al., 2015*), denoted as StaticMap in the ablation study. Since there is also relation-specific hyperplane strategy (*Wang et al., 2014*), and the relation-specific hyperplane can be regarded as a special case of relation-specific space, and the hyperplane projection is also a special case of mapping, therefore we also compare this relation-specific hyperplane strategy in the ablation study, denoted as PlaneMap. Moreover, we also include the representation without relation-specific consideration (*Bordes et al., 2013*) in the ablation study. In addition, the metric of distance in gene space representation is set to L1-norm in our approach, the metric can also be set to L2-norm in the ablation study. In the results of ablation study, the ROC curve demonstrates that for both STRING and iRefIndex interactions, the full version of our approach can yield better performance than those of the other cases (Fig. 4). Also, our approach with sparsity-inducing L1-norm also outperforms that with Euclidean L2-norm (Fig. 4). Generally, the full version of our approach with both dynamic mapping gene space and sparsity-inducing metric shows superiority to the other versions.

## Results of subtype-specificity visualization

In addition to the aforementioned experiments on cancer driver discovery, subtype-specificity of the investigated driver is also another important aspect in precision medicine of cancer diagnosis. Unfortunately, unlike the cancer driver genes, subtype-specificity of genes do not have benchmarking. Accordingly, we perform a visualization experiment of subtype-specificity indication of our approach as an intuitive and visualized way (*Xi et al., 2020*). In our approach, we can adopt the recovery loss score on the triplet (gene $G$, belong to $B$, subtype $S$) and obtain the loss score for subtype $S$. By switching the triplet tail across all subtypes in the triplet, we can compute the scores corresponding to the investigated gene across all the subtypes. Finally, we selected the subtype with minimizing recovery loss as the subtype-specificity of the gene to be tested.

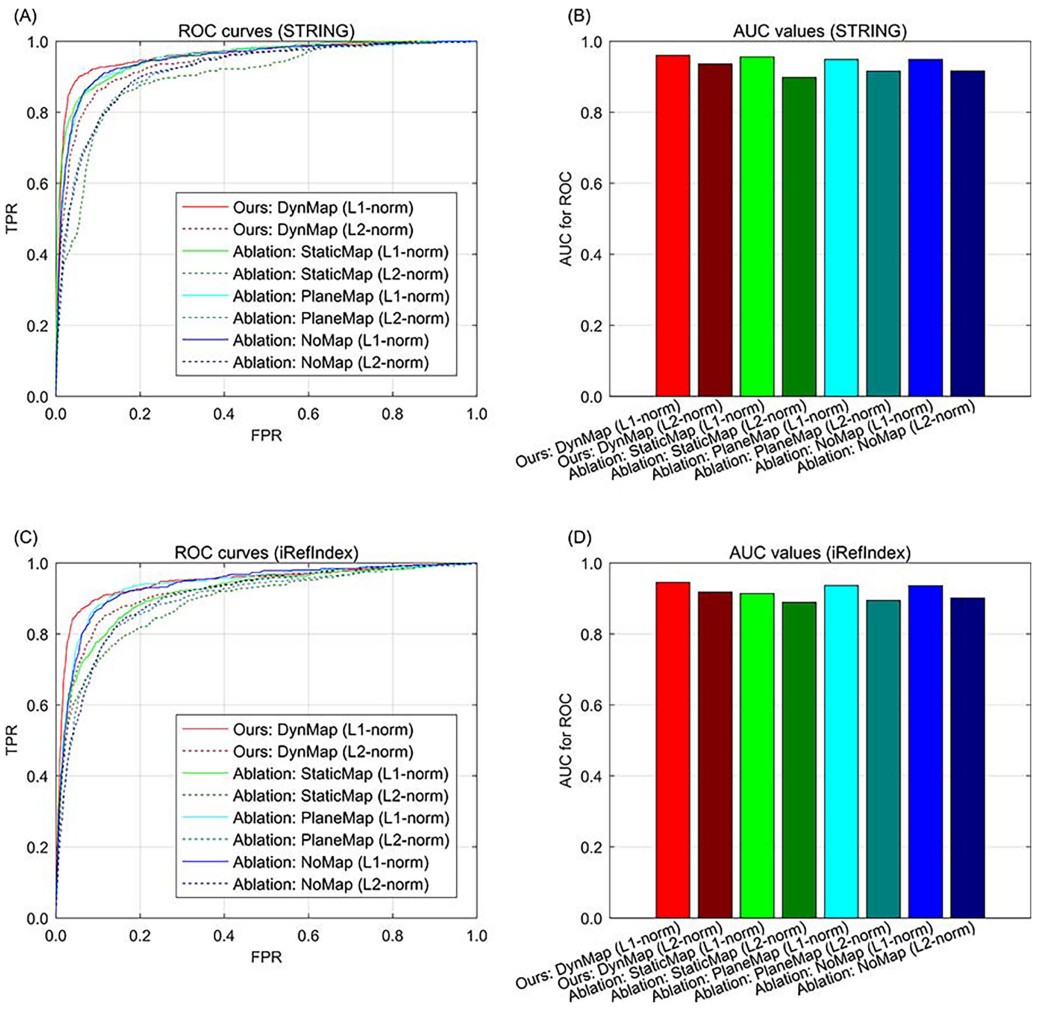

**Fig. 4** The ROC curves of our approach to ablation study in methodology on breast cancer data and gene interactions. (A) ROC curves for ablation study in methodology with interaction source of STRING. (B) AUC values for ablation study in methodology with interaction source of STRING. (C) ROC curves for ablation study in methodology with interaction source of iRefIndex. (D) AUC values for ablation study in methodology with interaction source of iRefIndex.

Full-size ⬛ DOI: 10.1093/bioinformatics/btu811

For discovered driver genes of our approach in the five-fold cross validation, we randomly select a fold and display the subtype-specificities of these genes. Since the dimension of gene space is usually larger than three and thus is difficult to display, we utilize the t-distributed neighbor embedding (t-SNE) on our gene space representation (*Van der Maaten & Hinton, 2008*; *Li et al., 2017*), and reduce the space dimension to 3D (Fig. 5). In the visualization of genes in the 3D space, we illustrate the distribution trends of different breast cancer subtypes through different colors. For providing a more comprehensive illustration on the details in Fig. 5, we also provide the three view drawings of the t-SNE 3D visualization for subtype-specificities of discovered drivers by our approach in Fig. S1, including the front view, the top view, and the side view of the 3D
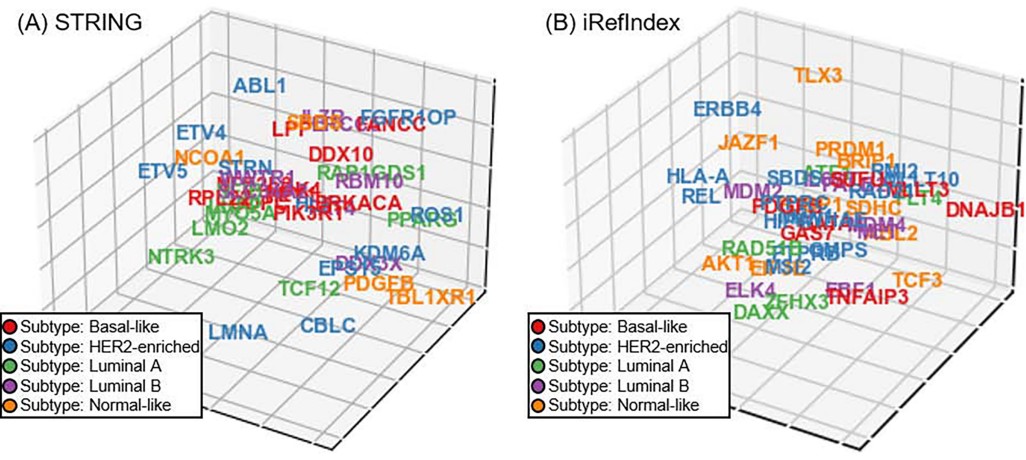

**Fig. 5 The t-SNE 3D visualization for subtype-specificities of discovered drivers by our approach. (A) Visualization result with interaction source of STRING. (B) Visualization result with interaction source of iRefIndex.** Full-size ◰ DOI: 10.1093/bioinformatics/btu811

visualization. As shown in Figs. 5A and 5B and Fig. S1, for the situations of both STRING and iRefIndex, we can observe distinct trends of each subtype (each color) across the space.

## Results of unscrambling functions enrichment

Although the aforementioned experiments have demonstrated the results of both cancer driver discovery and subtype-specificity indication of the discovered potential driver genes, the biological functions of these genes are still not unscrambled. Hence, we further apply the functional enrichment analysis on our discovered potential driver genes to find which biological functions are related to these genes. Here we adopt the functional enrichment analysis through a widely-used platform called Database for Annotation, Visualization and Integrated Discovery (DAVID) (*Sherman et al., 2022*). Since the top scored genes are expected to have more potential of being cancer drivers, we select the top 100 discovered genes into the functional enrichment analysis. The functional enrichment analysis yields the enriched functional terms of curated by the Kyoto Encyclopedia of Genes and Genomes (KEGG) (*Kanehisa et al., 2004*). Table 1 shows the top twenty functional terms unscrambled by the enrichment analysis. Through the unscrambling results, we can find that the top enriched functional terms are highly associated with cancer process, such as pathways in cancer, EGFR tyrosine kinase inhibitor resistance, JAK-STAT signaling pathway, cell cycle, microRNAs in cancer, transcriptional misregulation in cancer, and breast cancer. Generally speaking, by enrichment analysis Unscrambling, the discover potential driver genes of our approach are observed to be functionally associated with cancer process.

## Results of identified driver candidates

As for the novel driver candidates surprisingly in top hits, we also investigate the identified new drivers that are surprisingly in top hits and never been identified in other existing methods. It should be noted that, in our experiments, the true positives are identified genes that are also included in benchmarking database (*Sondka et al., 2018*), and the false

**Table 1 The functional enrichment analysis results of the discovered drivers of our approach with interaction sources of STRING and iRefIndex respectively.**

| Term (Results of STRING) | Percentage | p-value | FDR |
|---|---|---|---|
| hsa05200:Pathways in cancer | 27.90697674 | 2.10E−13 | 2.18E−11 |
| hsa05210:Colorectal cancer | 13.95348837 | 1.14E−11 | 5.95E−10 |
| hsa05215:Prostate cancer | 13.95348837 | 4.39E−11 | 1.52E−09 |
| hsa05205:Proteoglycans in cancer | 16.27906977 | 1.00E−09 | 2.60E−08 |
| hsa05213:Endometrial cancer | 10.46511628 | 4.97E−09 | 1.03E−07 |
| hsa05225:Hepatocellular carcinoma | 13.95348837 | 1.64E−08 | 2.85E−07 |
| hsa05220:Chronic myeloid leukemia | 10.46511628 | 4.44E−08 | 6.59E−07 |
| hsa01521:EGFR tyrosine kinase inhibitor resistance | 10.46511628 | 6.04E−08 | 7.26E−07 |
| hsa05226:Gastric cancer | 12.79069767 | 6.28E−08 | 7.26E−07 |
| hsa05216:Thyroid cancer | 8.139534884 | 1.74E−07 | 1.81E−06 |
| hsa05163:Human cytomegalovirus infection | 13.95348837 | 3.30E−07 | 3.02E−06 |
| hsa05221:Acute myeloid leukemia | 9.302325581 | 3.48E−07 | 3.02E−06 |
| hsa05218:Melanoma | 9.302325581 | 5.74E−07 | 4.26E−06 |
| hsa05223:Non-small cell lung cancer | 9.302325581 | 5.74E−07 | 4.26E−06 |
| hsa05224:Breast cancer | 11.62790698 | 6.71E−07 | 4.61E−06 |
| hsa05202:Transcriptional misregulation in cancer | 12.79069767 | 7.10E−07 | 4.61E−06 |
| hsa05214:Glioma | 9.302325581 | 7.61E−07 | 4.65E−06 |
| hsa05212:Pancreatic cancer | 9.302325581 | 8.33E−07 | 4.82E−06 |
| hsa05206:MicroRNAs in cancer | 15.11627907 | 1.11E−06 | 6.05E−06 |
| Term (Results of iRefIndex) | Percentage | p-value | FDR |
| hsa05200:Pathways in cancer | 23.65591398 | 5.59E−12 | 7.38E−10 |
| hsa05215:Prostate cancer | 11.82795699 | 5.44E−10 | 3.59E−08 |
| hsa05220:Chronic myeloid leukemia | 9.677419355 | 2.86E−08 | 9.43E−07 |
| hsa05212:Pancreatic cancer | 9.677419355 | 2.86E−08 | 9.43E−07 |
| hsa05225:Hepatocellular carcinoma | 11.82795699 | 1.14E−07 | 3.01E−06 |
| hsa04068:FoxO signaling pathway | 10.75268817 | 1.55E−07 | 3.40E−06 |
| hsa05223:Non-small cell lung cancer | 8.602150538 | 3.93E−07 | 7.40E−06 |
| hsa04630:JAK-STAT signaling pathway | 10.75268817 | 9.45E−07 | 1.56E−05 |
| hsa05210:Colorectal cancer | 8.602150538 | 1.33E−06 | 1.82E−05 |
| hsa04110:Cell cycle | 9.677419355 | 1.48E−06 | 1.82E−05 |
| hsa05166:Human T-cell leukemia virus 1 infection | 11.82795699 | 1.52E−06 | 1.82E−05 |
| hsa05224:Breast cancer | 9.677419355 | 4.72E−06 | 5.19E−05 |
| hsa05226:Gastric cancer | 9.677419355 | 5.21E−06 | 5.30E−05 |
| hsa05218:Melanoma | 7.52688172 | 7.04E−06 | 6.64E−05 |
| hsa05214:Glioma | 7.52688172 | 8.94E−06 | 7.87E−05 |
| hsa05161:Hepatitis B | 9.677419355 | 9.68E−06 | 7.99E−05 |
| hsa04919:Thyroid hormone signaling pathway | 8.602150538 | 1.31E−05 | 1.02E−04 |
| hsa05206:MicroRNAs in cancer | 11.82795699 | 2.92E−05 | 2.14E−04 |
| hsa05202:Transcriptional misregulation in cancer | 9.677419355 | 3.46E−05 | 2.37E−04 |
positives are identified genes that are not collected by benchmarking database (*Sondka et al., 2018*). In consideration of the lack of non-driver database, therefore the non-collected false positive genes identified by our model can also be regarded as potential new drivers. Especially, the driver gene candidates in top hits show more potential to be cancer drivers, such as the top ranked driver candidates C15orf65, ARNTL, and BCL5, which are not collected by benchmarking database (*Sondka et al., 2018*). Consequently, we demonstrate the top hit genes generated by our model as the new driver candidates, and the full list of the top ranked candidates, *i.e.*, false positives, are in Table S1. Generally, our identified results contain considerable numbers of novel drivers, and we welcome the wet-lab validation on these candidates.

## DISCUSSION

Discovering subtype-specific drivers is an inevitable demand in breast cancer precision medicine. The existing widely-used computational tools for driver discovery mainly focus on exploiting the information from DNA aberrations or gene interactions. However, recent studies have demonstrated that expect DNA aberrations, RNA alternations also play important roles in cancer driver events (*Calabrese et al., 2020*), but there is still a lack of an integration strategy for multiple types of aberrations from both DNA and RNA. Generally, there are mainly two reasons make the integration as a challenging task. The first reason is the data format incompatibility, where the data formats of different types of aberrations are distinct from each other. The second reason is the aberration type heterogeneity, where the patterns of multi-type aberrations also vary from each other. In this article, we propose an integration strategy for subtype-specific breast cancer driver discovery with a "splicing-and-fusing" framework. In our framework, to address the data format incompatibility, the "splicing-step" employs a knowledge graph structure to connect multi-type aberrations from the DNA and RNA data into a unified formation (*Nickel et al., 2015*). To tackle the aberration type heterogeneity, the "fusing-step" adopts a dynamic mapping gene space integration approach to represent the multi-type information by vectorized profiles (*Wang et al., 2017*). The evaluation experiments also demonstrate the advantages of our approach in both the integration of multi-type aberrations from DNA and RNA and the discovery of subtype-specific breast cancer drivers.

The main advantages of our study can be summarized as three points. The first advantage is that the multi-type aberrations can contain more information beyond the scope of breast cancer genomic data. By adding the RNA alternations, interactions between genes, the subtype annotations of samples, and the supervision information of known benchmarking drivers, the discovery task of subtype-specific cancer drivers has a richer connotation than those of most of the existing computational tools. The second advantage is that the knowledge graph structure can successfully overcome the issue of data format incompatibility. We can redescribe the existing data with a series of triplets of facts, and splice them together by connecting common entities and relations into an integrated knowledge graph of multi-type aberrations of breast cancer. The third advantage is that the idea of dynamic mapping strategy can fuse the triplet data into relation-specific gene space, and the low-dimensional fused gene space can describe the aberration type heterogeneity

as many-sided representations. Through the advantages above, our approach of multi-type aberration integration can not only discover potential driver genes from breast cancer data, but also indicate the subtype-specificities of these potential drivers by the integration of both DNA and RNA data of breast cancer samples.

With the advancement of artificial intelligence in biomedical research (*Xi et al., 2021a*; *Liu, Wang & Xi, 2022*) and many other fields (*Chougule et al., 2022*), many related works of integration of DNA and RNA have emerged for cancer samples. In early studies, integrating DNA and RNA information had already been explored in low-purity mutation detection (*Wilkerson et al., 2014*). There is also previous work that extends beyond the driver gene identification, achieving the driver modules of multi-omic data (*Silverbush et al., 2019*). To further identify driver aberrations from detected mutations in a cohort of samples, (*Siegel et al., 2018*). adopt the integration of DNA and RNA to analyze early drivers of metastatic breast cancer through one of the comparison methods DawnRank (*Hou & Ma, 2014*) in our experiment on their collected dataset. *Zhang et al. (2022)* proposes an integration of two-types of aberration for identifying early drivers involved in metastasis of gastric cancer by statistical analysis (*Zhang et al., 2022*), in comparison, our proposed model can regard the aberration types at fine-grained level. Also, to detect drivers of endocrine resistance in estrogen receptor–positive breast cancer, *Xia et al. (2022)* integrates DNA and RNA aberrations by linear mixed model, where the different types of aberrations are processed as unified variables. In contrast, our proposed data integration based on knowledge graph (*Xi et al., 2023*) can keep the information of the aberration type heterogeneity, and avoid the curse of dimensionality at the same time.

Despite the achievements of our integrated approach, there are also some limitations in our study. An inevitable limitation is that our approach relies on a strong restriction of the alignment of the DNA and RNA data samples. If some breast cancer samples only contain DNA data or RNA data, these samples cannot be used in our integration analysis (*Ebrahim et al., 2016*). Actually, there are less than one thousand cancer samples that satisfy the requirement of combination of both DNA and RNA data in the TCGA database used in our study. Also, since the gene space representations are expected to be information preserving, the learning process of these representations requires more computational resources than those of the traditional statistical based methods. As for the future plan of this study, a promising future work of our study is to expand the framework to the pan-cancer analysis (*Leiserson et al., 2015*). In consideration of the capability of the multi-type aberration fusion and the subtype-specificity indication, the integration of various cancer types in pan-cancer analysis is also feasible in some extent. In summary, we design a "splicing-and-fusing" framework with knowledge graph connection and dynamic mapping gene space fusion of multi-type aberrations data from DNA and RNA, and the application of our approach on breast cancer samples can successfully discover potential cancer drivers with subtype-specificity indication.

## ACKNOWLEDGEMENTS

We would like to thank Dr. Liying Yang for her helpful suggestions.

## Funding

This work is supported by the National Natural Science Foundation of China (Grant Nos. 61901322, 62202117, and 61974109). This work is also supported by the Special Foundation in Department of Higher Education of Guangdong (Grant No. 2022ZDX2053). The funders had no role in study design, data collection and analysis, decision to publish, or preparation of the manuscript.

## Grant Disclosures

The following grant information was disclosed by the authors:
National Natural Science Foundation of China: 61901322, 62202117, and 61974109.
Special Foundation in Department of Higher Education of Guangdong: 2022ZDX2053.

## Competing Interests

The authors declare that they have no competing interests.

## Author Contributions

- Jianing Xi conceived and designed the experiments, performed the experiments, analyzed the data, prepared figures and/or tables, authored or reviewed drafts of the article, and approved the final draft.
- Zhen Deng conceived and designed the experiments, performed the experiments, analyzed the data, prepared figures and/or tables, and approved the final draft.
- Yang Liu analyzed the data, prepared figures and/or tables, authored or reviewed drafts of the article, and approved the final draft.
- Qian Wang performed the experiments, analyzed the data, prepared figures and/or tables, and approved the final draft.
- Wen Shi conceived and designed the experiments, authored or reviewed drafts of the article, and approved the final draft.

## Data Availability

The data is available at UCSC Xena: https://xenabrowser.net/.

## Supplemental Information

Supplemental information for this article can be found online at http://dx.doi.org/10.7717/peerj.14843#supplemental-information.

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
