# Peer review of "Integrating multi-type aberrations from DNA and RNA through dynamic mapping gene space for subtype-specific breast cancer driver discovery"

_PeerJ, doi:10.7717/peerj.14843_

## Round 0.1 · original submission · Minor Revisions

Please address the concerns of all reviewers and revise the manuscript accordingly.

Reviewer 1 ·

Basic reporting

This is an interesting manuscript to present a potential useful tool to discover more breast cancer drivers.
They showed the schematic diagram of “splicing and fusing” framework including (A) Data acquisition of breast cancer samples, (B) “splicing-step”, (C) ”fusing-step” and (D) ”Discovering-step”. In my opinion, the authors should show some examples to verify they can identify the potential drivers to support their claim. Importantly, authors should open this program to the community.

Experimental design

This is an original primary research.

Validity of the findings

Novelty is good but have not idea about the impact. Authors should allow more researchers to use their method to calculate the drivers.

Additional comments

no

·

Basic reporting

Overall, the manuscript is nicely written. The introduction and background clearly showed the context. Similarly, the discussion touches on some limitations of the study, related work in the field and highlights importance of the used approach.

Suggestions:

1. The font size used in Fig.2 (A-D), Fig.3 (A-D) and Fig.4 (A-D) is not readable. I would suggest authors to use larger fonts for the titles, axis labels, legends, and tick labels.
2. In the discussion, more related work if it is available needs to be cited.
3. On line 143: ‘gene aberrations from DNA’ is mentioned two times. As per my understanding it should be RNA abnormality or RNA aberrations.
4. Kindly check the double inverted commas used throughout the manuscript.
5. On line 461: Figure 3B should be replaced by 5B.

Experimental design

Authors performed a unique study where aberrations from both DNA and RNA were unified to discover subtype specific breast cancer drivers. Using a ‘splicing-and-fusing’ framework helped the research team to overcome the mentioned challenges such as data format incompatibility and the aberration type heterogeneity. Experimental design is scientifically sound, and the required details are provided in methods and results.

Validity of the findings

The rationale and benefit to literature is clearly stated. Similarly, conclusions are well stated in the manuscript.
However, I would suggest authors to use different subtitles to describe results. In the current manuscript, the result subtitles look like the subtitles used in methods. Kindly use subtitles which are more informative or give readers an idea about what is written in the following paragraph.

Reviewer 3 ·

Basic reporting

Xi et al., have mostly used in silico approaches to find DNA and RNA aberrations for driver event discovery of cancer. The study seems ambitious and potentially be expanded in the discovery of cancer drivers other than breast cancer. Few things needs to be addressed to have the manuscript in an acceptable form.
1.) Figure 1 looks complicated and it cannot be interpreted easily (especially C and D)
2.) The text in all other figures is not clear. I suggest authors to make it bold and increase the font
3.) in Figure 5 there are many drivers that a reader cannot see. Authors should find a better way of showing it
4.) Needs extensive correction of English

Experimental design

It is not clear to me if the DNA and RNA aberrations are identified from same breast cancer sample. Wouldn't be ideal to use the same sample so the driver identified will have low false discovery rate?

Validity of the findings

1.) Authors will also be interested to know if the particular methodology identified any new drivers that are surprisingly in top hits and never been identified in other methodologies used by researchers
2.) It would also be nice to know the false positives generated in this methodology.

Additional comments

I suggest authors to cite articles in a standard way rather than having them in continuation of a sentence.

---

## Round 0.2 · accepted · Accept

Thank you very much for carefully addressing all the issues pointed out by the reviewers and for the thorough revision. Since all concerns were adequately addressed, I am pleased to accept the revised manuscript.